# Manifestation of Pathology in Animal Models of Diabetic Retinopathy Is Delayed from the Onset of Diabetes

**DOI:** 10.3390/ijms25031610

**Published:** 2024-01-28

**Authors:** Samuel Cubillos, Andrius Kazlauskas

**Affiliations:** 1University of Illinois at Chicago, College of Medicine, Chicago, IL 60612, USA; scubil2@uic.edu; 2Lions Illinois Eye Research Institute, Chicago, IL 60612, USA

**Keywords:** diabetic retinopathy, resistance to diabetic retinopathy, mouse models, protection, mitophagy, hyperglycemia-induced mitochondrial adaptation

## Abstract

Diabetic retinopathy (DR) is the most common complication that develops in patients with diabetes mellitus (DM) and is the leading cause of blindness worldwide. Fortunately, sight-threatening forms of DR develop only after several decades of DM. This well-documented resilience to DR suggests that the retina is capable of protecting itself from DM-related damage and also that accumulation of such damage occurs only after deterioration of this resilience. Despite the enormous translational significance of this phenomenon, very little is known regarding the nature of resilience to DR. Rodent models of DR have been used extensively to study the nature of the DM-induced damage, i.e., cardinal features of DR. Many of these same animal models can be used to investigate resilience because DR is delayed from the onset of DM by several weeks or months. The purpose of this review is to provide a comprehensive overview of the literature describing the use of rodent models of DR in type-1 and type-2 diabetic animals, which most clearly document the delay between the onset of DM and the appearance of DR. These readily available experimental settings can be used to advance our current understanding of resilience to DR and thereby identify biomarkers and targets for novel, prevention-based approaches to manage patients at risk for developing DR.

## 1. Introduction

Diabetic retinopathy (DR) is a chronic sequelae of diabetes mellitus (DM) and a leading cause of blindness worldwide [1,2,3]. Early detection of DR symptoms is crucial for timely intervention and prevention of irreversible vision loss [2]. In humans, the diagnosis of DR is based exclusively on the presence of retinal vascular abnormalities such as microaneurysms, hemorrhages, exudates, leakage resulting in edema, and neovascularization [1,2,3,4]. Although neural and visual abnormalities also develop, they are not considered in the diagnosis [1,2].

In contrast, all types of retinal dysfunction (vascular, neural, and visual) are used to assess DR in animal models [4]. Common pathologic features of DR in rodents affect both the vascular and neural compartments of the retina. Vascular abnormalities include inflammation of the endothelium, increased retinal vascular permeability, and manifestation of acellular capillaries. Within the neural retina, diagnostic features of DR include the death of various neural cell types (thinning of the retina), reactive gliosis, and neural dysfunction. These outcomes are typically detected using immunofluorescence, immunohistochemical staining, electroretinogram (ERG), and optical coherence tomography (OCT). While rodent models develop the early stages of DR, they do not develop the advanced, proliferative stage of DR, which involves neovascularization of the retina. Retinal neovascularization can be modeled with the oxygen-induced retinopathy protocol [5]; however, animals are not DM in this experimental setting and, hence, constitute a major difference from the clinical scenario. Thus, DM induces the early stages of retinal damage in experimental animals, similar to what occurs in humans.

Rodent models are some of the most useful models for studying the pathology of DR. Not only does the pathogenesis closely mimic the retinopathy-free period and then initial phases of DR in a fraction of the time (Figure 1), but rodents are also extraordinarily versatile when it comes to gene modification. Rodent models have been and continue to be instrumental in documenting the duration of DM necessary for first detecting DM-related damage in both the vascular and neural compartments of the retina, and the progressive nature of such damage as the duration of DM is prolonged. Such descriptive studies are the bedrock for both mechanistic studies seeking to understand the nature of DM-induced damage as well as translational efforts seeking to identify agents that prevent and even reverse such damage. This large body of the literature, which is focused on DR, i.e., damage to the retina that becomes detectable after weeks or months of DM and progresses as the duration of DM is prolonged, only indirectly addresses resilience to DR.

Recent studies focused on resilience to DR have provided additional evidence for its existence and have begun to elucidate the underlying mechanism [6,7]. In both streptozotocin-induced type 1 diabetic (T1D) and db/db genetically modified type 2 diabetic (T2D) mice, durations of DM that were insufficient to cause detectable damage increased expression of NAD(P)H Quinone Dehydrogenase 1, Sod2, Gclc, and other antioxidant defense genes within the retina [7]. Furthermore, the retinal vessels of these DM mice became resistant to oxidative stress-induced death. As the duration of DM increased, resilience was lost—the expression of antioxidant defense genes declined; the vasculature became more (instead of less) vulnerable to death, and both neural and vascular hallmarks of DM appeared. Thus, the delay in developing DR corresponded to a period of resilience to DM-associated damage.

**Figure 1 ijms-25-01610-f001:**
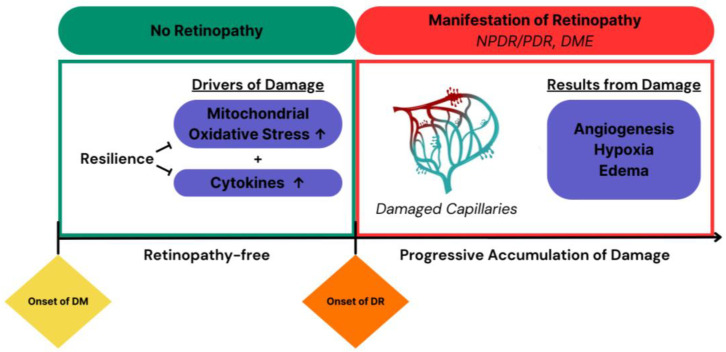
DR pathogenesis involves loss of resilience, which is a prerequisite for the subsequent accumulation of damage within the retina. In both humans and experimental animals, a retinopathy-free period precedes the appearance of DR. Recent discoveries in both T1D and T2D mice indicate that the retinopathy-free period is associated with enhanced tolerance of insults that cause damage to the retina, such as increased mitochondrial oxidative stress and cytokine production [7]. Loss of resilience sets the stage for progressive accumulation of retinal damage, which indicates the existence of DR.

Of the two commonly used rodent models of DR, mice are used more often than rats. While the eye of a rat is larger than a mouse’s eye, the lower cost and greater selection of genetically modified mice are some of the reasons why mice are more popular than rats in the field of DR research.

In this review, we provide a comprehensive overview of the current literature demonstrating that DR does not develop coincident with the onset of DM in murine and rat models of T1D and T2D. Rather, DR becomes detectable after weeks or months of DM, whereupon its severity increases as the duration of DM is prolonged. The progressive nature of DR suggests the existence of resilience, which protects from and/or repairs DM-induced damage, and that deterioration of such systems is an essential component of DR pathogenesis.

## 2. The Literature Review

### 2.1. Murine Models of Type-1 Diabetic-Induced Retinopathy

T1D rodent models mimic the autoimmune destruction of pancreatic β-cells, leading to insulin deficiency. Mouse models such as the streptozotocin (STZ)-induced diabetic mouse and rat, non-obese diabetic (NOD) mouse, and alloxan-induced mouse models have been extensively employed in investigations of DR. STZ and alloxan are chemical compounds commonly used to induce T1D in mice through intravenous or intraperitoneal injection. Although the exact mechanism behind alloxan’s effects is not fully understood, both of these chemicals preferentially affect the pancreatic β-cells due to the cells’ elevated levels of the GLUT2 glucose transporter. STZ and alloxan enter β-cells through GLUT2 and are metabolized into toxic side-products that damage DNA, cause oxidative stress, and ultimately lead to the death of these insulin-producing cells. At a concentration of STZ that kills Glut2-expressing cells, it is non-cytotoxic toward cells that do not express its transporter [8]. Multiple injections of moderate doses of STZ cause gradual death of beta cells that results from the combined action of STZ-mediated cytotoxicity (alkylating DNA) and immune attack [9,10,11]. It is this destruction of insulin-producing cells by which STZ induces DM; insulin supplementation or a pancreatic islet transplant prevents DM in STZ-treated animals [12]. Similarly, the breakdown of the blood-retinal barrier does not occur in STZ-treated animals that are given insulin [12,13,14]. While STZ has effects unrelated to eliminating the insulin-producing cells, such side effects are more common in single, high-dose STZ protocol that also induces DM [15]. NOD mice, on the other hand, exhibit T1D-like phenotypes due to the interplay of several factors that result in genetic susceptibility, autoimmune dysregulation, inflammatory infiltration, and environmental triggers that lead to the destruction of pancreatic β-cells [4,16]. Unlike the way DR is diagnosed in humans (based exclusively on vascular parameters), the presence of DR in T1D is evaluated considering vascular, neural, and visual dysfunction. The earliest signs of DR in T1D mice have been reported as early as one week after DM onset, as can be seen in Table 1.

Throughout the literature on T1D mouse models, there has been consistent notice of a delay between the onset of DM and outcomes seen in DR, although of variable length. Once these outcomes have developed, they persist and intensify as DM and DR continue to progress, with the exception of leukostasis and inflammation [17]. Consequently, the longer the animal has DM, the greater the differences will be between the DM and non-DM groups. For this reason, investigators focused on DM-induced damage of the retina, prolonging the duration of DM until after resilience had deteriorated. The earliest signs of vascular dysfunction were seen following one week of DM and consisted of degenerated pericytes, acellular capillaries, and minor vessel proliferation [18,19,20,21,22,23,24,25,26,27,28,29]. Neural dysfunction, which was measured with electroretinogram (ERG), was absent after 1 week of DM and detectable two weeks after the onset of DM.

**Table 1 ijms-25-01610-t001:** Articles discussing Type-1 diabetic mouse models.

Authors, (Year)	Model	Collected Tissue	Mechanism of Analysis in Diabetic Retinopathy	Monitoring of DR Progression	Earliest Detection of Diabetic Retinopathy	Result/Outcome
Weerasekera, L.Y., Balmer, L.A., Ram, R., et al. (2015) [18]	STZ Alloxan	Retina	Flat-mount whole retina Microscopy Microarray Retinal trypsin digest	Yes	1 week of diabetes	Mice retinas showed degenerated pericytes, acellular capillaries, minor vascular proliferation, gliosis of Müller cells, and loss of ganglion cells (GC).
Muriach, M., Bosch-Morell, F., Alexander, G., et al. (2006) [30]	Alloxan	Retina	Oxidative stress markers and functional tests ERG Morris water maze test	None	2 weeks of diabetes	Mice retinas showed that ocular malondialdehyde (MDA) concentration was higher than controls, whereas GSH concentration and GPx activity decreased in the diabetic retina. ERG B-wave amplitude decreased in diabetic animals compared to controls.
Lee, Y.J., Jung, S.H., Hwang, J., et al. (2017) [31]	STZ	Retina	Fluorescein angiography Confocal microscopy ELISA	None	2 weeks of diabetes	Mice retinas showed higher levels of extravasation of FITC-dextran, which were observed in the retinas of diabetic mice. ROS levels were also elevated in diabetic compared to normal mouse retinas.
Leal, E.C., Manivannan, A., Hosoya, K., et al. (2007) [32]	STZ	Retina	Flat-mount microscopy Confocal microscopy Evans Blue Western blot Flow cytometry Immunohistochemistry	Yes	2 weeks of diabetes	Mice retinas showed blood-retinal barrier (BRB) breakdown, leukocyte number, ICAM-1 protein levels, eNOS levels, and Nitrotyrosine immunostaining, which increased compared to controls.
He, J., Wang, H., Liu, Y., et al. (2015) [33]	STZ Ins2Akita	Retina	Western blot Immunofluorescence RT-PCR	None	2 weeks of diabetes	Mice retinas showed significantly increased numbers of leukocytes adhering to retinal vasculature compared to controls.
Sergeys, J., Etienne, I., Van Hove, I., et al. (2019) [17]	STZ	Retina	Visual acuity Contrast sensitivity ERG SD-OCT Leukostasis Immunohistochemistry FITC-BSA	Yes	2 weeks of diabetes	Mice retinas showed a rapid increase in leukostasis retinal vasculature compared to controls.
Feit-Leichman, R.A., Kinouchi, R., Takeda, M., et al. (2005) [27]	STZ	Retina	TUNEL assay Western blot Photomicrography Trypsin-digest Transmission electron microscopy	Yes	2 weeks of diabetes	Mice retinas showed an increase in TUNEL-positive cells in the GCL compared to controls.
Lee, S., Harris, N.R. (2008) [34]	NOD	Retina	Retinal arteriolar diameter analysis RBC velocities	None	3 weeks of diabetes	Hyperglycemic non-obese diabetic (NOD) mice retinas demonstrated retinal arterioles closely paired with venules that were roughly 16% more constricted, and flow was roughly 33% reduced compared to normoglycemic NOD controls.
Rangasamy, S., McGuire, P.G., Franco Nitta, C., et al. (2014) [20]	STZ	Retina	Cell Culture RT-qPCR ELISA Confocal microscopy Flow Cytometry Electrical Cell-Substrate Impedance Sensing (ECIS) Quantitative Assessment of BRB Permeability	None	4 weeks of diabetes	Mice retinas showed numerous GFP+ round cells (monocytes/macrophages), without processes, present within the retinal tissue and outside the vasculature.
Sasaki, M., Ozawa, Y., Kurihara, T., et al. (2010) [35]	STZ	Retina	ERG Immunoblotting Histologic analysis TUNEL and caspase-3 staining	None	4 weeks of diabetes	Mice retinas demonstrated DHE staining in all retinal layers with decreased ERG amplitude of OPs reflecting inner retinal functional status. Synaptophysin protein and BDNF were decreased, while ERK activation was elevated compared to controls.
Kurihara, T., Ozawa, Y., Shinoda, K., et al. (2006) [36]	STZ	Retina	Immunoblotting RT-qPCR ERG	None	4 weeks of diabetes	Mice retinas showed increased retinal levels of angiotensin II and its receptor AT1R, enhanced ERK phosphorylation, prolonged implicit time of OP2 and OP3, and lower amplitude OP3 compared to controls.
Li, C.-R., Sun, S.-G. (2010) [19]	NOD	Retina	ELISA Transmission electron microscopy	Yes	4 weeks of diabetes	Female NOD mice retinas demonstrated a significantly increased membrane thickness, microvascular occlusion, and perivascular edema in retinal capillary basement, chromatin margination with nuclear deformation in the endothelial and pericyte cells, and an irregular aggregation of nuclear chromatin with vacuolar degeneration in the ganglion cells compared to controls.
Wang, Q., Navitskaya, S., Chakravarthy, H., et al. (2016) [22]	STZ	Retina	RT-qPCR TaqMan miRNA Assays Confocal microscopy Fluorescence microscopy	None	4 weeks of diabetes	Mice retinas showed increased vascular permeability in the retina compared to controls.
Miloudi, K., Oubaha, M., Ménard, C., et al. (2019) [37]	STZ	Retina	RT-qPCR Western Blot	Yes	4 weeks of diabetes	Mice retinas demonstrated increased levels of Jag1, DLL4, and Notch1 transcripts compared to controls.
Yoon, C.H., Choi, Y.E., Cha, Y.R., et al. (2016) [21]	STZ	Retina	Immunofluorescence Confocal microscopy TUNEL assay Flow cytometry RT-qPCR In-vitro angiogenesis	None	4 weeks of diabetes	Mice retinas showed decreased capillary area/retinal area ratio, increased number of acellular capillaries, and decreased capillary diameter compared to controls.
Chung, Y.R., Choi, J.A., Koh, J.Y., et al. (2017) [38]	STZ	Retina	Fluorescein angiography FITC staining Immunohistochemistry Western blot	None	4 weeks of diabetes	Mice retinas showed increased expression levels of GRP78, pPERK, peIF2α, MCP-1, cytokines, and TNFα in retinal tissues compared to controls.
Barber, A.J., Antonetti, D.A., Kern, T.S., et al. (2005) [39]	Ins2Akita	Retina	GFAP staining Immunoreactivity	Yes	4 weeks of diabetes	Hyperglycemic Ins2Akita mice retinas showed increased apoptosis identified by immunoreactivity for active caspase-3 compared to controls.
Kumar, S., Zhuo, L. (2010) [40]	STZ	Retina	Confocal scanning laser ophthalmoscopic imaging Immunohistochemistry RT-qPCR	Yes	5 weeks of diabetes	Mice retinas showed increased GFAP-GFP transgene expression, average TFGI, and average number of astrocytic cell bodies compared to controls.
Sohn, E.H., Van Dijk, H.W., Jiao, C., et al. (2016) [41]	STZ Db/db	Retina	SD-OCT Immunohistochemistry Trypsin-digest DAPI nuclear counterstaining	Yes	6 weeks of diabetes	Mice retinas showed significant thinning of the nerve fiber layer (NFL) and GCL on OCT, and GC loss but no reduction in density compared to controls.
Nagai, N., Izumi-Nagai, K., Oike, Y., et al. (2007) [42]	STZ	Retina	RT-qPCR Western blot RT-qPCR ELISA	None	7 weeks of diabetes	Mice retinas showed increased mRNA expression of angiotensinogen, AT1-R, AT2-R, upregulated protein levels of angiotensin II, AT1-R, AT2-R, ICAM-1, VEGF, enhanced nuclear translocation of NF-κB p65, and MCP-1 compared to controls.
Zheng, L., Du, Y., Miller, C., et al. (2007) [43]	STZ	Retina	Fluorometric assay kit Western blot ELISA FITC Flat-mount and fluorescence microscopy Trypsin digest Immunohistochemical stains ERG	Yes	8 weeks of diabetes	Mice retinas showed an increase in eNOS production compared to controls.
Vincent, J.A., Mohr, S. (2007) [44]	STZ	Retina	Minocycline studies Tissue cultures Retinal and cell lysates Caspase activity assay Interleukin-1B ELISA Histological assessment of retinal vascular pathology	None	8 weeks of diabetes	Mice retinas showed significantly higher levels of caspase-1 activity and IL-1β levels compared to controls.
Mohr, S., Xi, X., Tang, J., et al. (2002) [45]	STZ	Retina	Western blot TUNEL assay	Yes	8 weeks of diabetes	Mice retinas demonstrated increased activity of several caspases -1, -2, -4, -5, -6, -8, and -9 compared to controls.
Kady, N.M., Liu, X., Lydic, T.A., et al. (2018) [23]	STZ	Retina	RT-qPCR Immunoblotting siRNA Western blot Immunocytochemistry Immunogold Electron Microscopy Mass Spectrometry	None	8 weeks of diabetes	Mice retinas showed significant increase in vascular permeability and downregulation of endothelial ELOVL4 expression compared to controls.
Saadane, A., Lessieur, E.M., Du, Y., et al. (2020) [46]	STZ	Retina	SD-OCT ERG Western blot Lucigenin assay	Yes	8 weeks of diabetes	Mice retinas showed increased vascular permeability and reduced ELOVL4 compared to controls.
McVicar, C.M., Ward, M., Colhoun, L.M., et al. (2015) [47]	STZ	Retina	RT-qPCR Confocal microscopy Fluorescent confocal microscopy Retinal flat mounts Epifluorescent microscopy	Yes	8 weeks of diabetes	Mice retinas showed accumulation of methylglyoxal (MG)-derived adducts and MG levels compared to controls.
Ouyang, H., Mei, X., Zhang, T., et al. (2018) [24]	STZ	Retina	Evans blue Retinal histology Western blot ELISA Immunofluorescence Immunohistochemical analysis	None	8 weeks of diabetes	Mice retinas showed increased leakage of Evans blue dye, decreased percentage of claudin-1 and -9, significant thinning of INL and ONL, increased percentage of iba1/actin ratio, increased nuclear expression of NFκBp65, increased mRNA expression of inflammatory cytokines, elevated levels and mRNA expression of VEGF compared to controls.
Liu, H., Tang, J., Du, Y., et al. (2015) [48]	STZ	Retina	Fluorescence microscopy Elastase digestion and PAS staining Photomicrography Immunoblotting Chromatography Virtual optokinetics	None	8 weeks of diabetes	Mice retinas showed increased superoxide generation and expression of inflammatory proteins compared to controls.
Du, Y., Cramer, M., Lee, C.A., et al. (2015) [49]	STZ	Retina	Spatial frequency threshold Contrast sensitivity	Yes	8 weeks of diabetes	Mice retinas showed decreased spatial frequency and increased superoxide production compared to controls.
Gaucher, D., Chiappore, J.A., Pâques, M., et al. (2007) [4]	Alloxan	Retina	Fluorescein Angiography ERG TUNEL assay Immunocytochemistry Scanner laser ophthalmoscope (SLO)	Yes	8 weeks of diabetes	Mice retinas showed increased superoxide, iNOS, and b-wave amplitude seen on ERG compared to controls.
Li, G., Veenstra, A.A., Talahalli, R.R., et al. (2012) [50]	STZ	Retina	FITC Immunohistochemistry RT-qPCR Coculture flow cytometry mREC luminescence Trypsin digest	None	8–10 weeks of DM	Mice retinas showed increased superoxide production, mRNA for iNOS, COX-2, TNF-α, ICAM-1, leukostasis, and dead mRECs compared to controls.
Gubitosi-Klug, R.A., Talahalli, R., Du, Y., et al. (2008) [51]	STZ	Retina	Trypsin digest FITC Fluorescence microscopy Western blot Immunohistochemistry	Yes	12 weeks of diabetes	Mice retinas showed increased leukostasis and superoxide production compared to controls.
Gastinger, M.J., Singh, R.S., Barber, A.J. (2006) [52]	Ins2Akita	Retina	TUNEL analysis Immunohistochemistry	None	12 weeks of diabetes	The Ins2Akita mice retinas showed 72% more TUNEL-positive nuclei, suggesting increased apoptosis, than controls.
Gastinger, M.J., Kunselman, A.R., Conboy, E.E., et al. (2008) [53]	Ins2Akita	Retina	Measuring density and morphology of retinal ganglion cells. Immunohistochemistry Confocal imaging Morphometric analysis via Sholl analysis	None	12 weeks of diabetes	Mice retinas showed loss of GC from the peripheral retina, morphologic changes in dendrites from surviving large ON-type cells, increased vascular permeability, microglial cell reactivity, leukostasis, caspase-3 activity, and reduced insulin receptor kinase activity compared to controls.
Shaw, S.G., Boden, J.P., Biecker, E., et al. (2006) [54]	NOD	Retina	Confocal microscopy RT-PCR	None	16 weeks of diabetes	NOD mice retinas showed regional microvascular degeneration and/or lack of perfusion in the retina with approximately 50% reduced vascular filling compared to wild-type controls.
Liu, Q., Zhang, F., Zhang, X., et al. (2018) [55]	STZ	Retina	Immunofluorescence staining Fluorescence microscopy Concanavalin A lectin perfusion Evans blue Western blot RT-qPCR	None	32 weeks of diabetes	Mice retinas showed mild to moderate reactive gliosis, upregulation of Nrf2 and NLRP3 inflammasome, enhanced ROS formation and CellROX fluorescence, increased leukostasis and vascular leakage, and increased expression of NLRP3, Caspase-1 p20, IL-1β p17, and ICAM-1 compared to controls.
Joussen, A.M., Poulaki, V., Le, M.L., et al. (2004) [28]	STZ	Retina	Evans blue Trypsin digest Electron microscopy	None	44 weeks of diabetes	Mice retinas showed increased number of adherent leukocytes, endothelial cell injury, and BRB breakdown compared to controls.

Once outcomes such as visual acuity decreases, reduced contrast sensitivity, ERG changes, NFL-GCL thinning, RGC and amacrine cell neurodegeneration, and gliosis have developed, they persist and intensify with the duration of DM, with the exception of leukostasis and inflammation [17]. Although a variety of markers have been used throughout the literature to study the role of inflammation in DR, some of the more common markers included an elevated number of leukocytes, leukocyte adhesion, microglial infiltration, production of cytokines, eNOS, and several caspases [20,32,33,38,43,44,45]. Every murine model of DR shows the progression of the disease, i.e., the extent and nature of the damage of the retina increases, but not the transition from NPDR to PDR.

### 2.2. Murine Models of Type-2 Diabetic-Induced Retinopathy

T2D rodent models mimic insulin resistance and impaired glucose metabolism. The T2D models rely on various genetic modifications that lead to DM and thus show a much wider variation between strains. The leptin receptor-deficient (db/db) mouse and high-sugar diet-induced models are some of the commonly used models in this realm [4,16]. Because the leptin receptor in db/db mice is non-functional, the leptin signaling pathway is disrupted and leads to dysregulation of appetite and satiety [4,16]. Such mice become obese, resistant to insulin, hyperinsulinemic, and develop pancreatic β-cell dysfunction that causes T2D symptoms. In a similar fashion, high-sugar diet-induced models of T2D are fed specific diets that lead to obesity and glucose dysregulation, causing overt diabetic phenotypes [4]. Db/db mice develop DM between 4–12 weeks of life, but commonly at 8 weeks. In contrast to the terminology found in the literature on T1D STZ-induced mouse models, the literature studying the db/db model uses “weeks of age” instead of time from DM onset. This is because the onset of DM is more difficult to pinpoint in this T2D mouse model as compared to the STZ-induced T1D models. In these scenarios, the approximate duration of DM is 8 weeks shorter than the age of the mice. The section earliest detection of diabetic retinopathy, of Table 2 depicts this shift in terminology.

Damage to the retina is time-sensitive, with prolonged DM resulting in worsening manifestations of DR without any worsening of DM, suggesting the progressive nature of DR. In T2D mice, the earliest indication of retinal dysfunction occurred at 10 weeks of age (after approximately 2 weeks of DM) and included signs of apoptotic changes in the INL, ONL, and GCL retina layers, along with increased vascular permeability and leakage [41,56,57,58].

**Table 2 ijms-25-01610-t002:** Articles of Note discussing Type-2 diabetic mouse models (db/db).

Authors, (Year)	Model	Collected Tissue	Mechanism of Analysis in Diabetic Retinopathy	Monitoring of DR Progression	Earliest Detection of Diabetic Retinopathy	Result/Outcome
Samuels, I.S., Bell, B.A., Pereira, A., et al. (2014) [59]	Db/db	Retina	ERG OCT	Yes	8 weeks of age (DM onset at 4 weeks of age)	Mice retinas showed reduced b-wave amplitudes compared to controls.
Bogdanov, P., Corraliza, L., Villena, J.A., et al. (2014) [60]	Db/db	Retina	ERG Transendothelial electrical resistance Mitophagy TBH challenge assay Seahorse analysis Western blot PCR analysis RNA sequencing LDH Fluorescence-activating cell sorting Glucose consumption rate Tissue culture	Yes	8 weeks of age	Mice retinas showed higher levels of glutamate, glial activation, TUNEL-positive cells, activated caspase-3, thickening in ONL and INL, and decreased number of cells in GCL compared to controls.
Hernández, C., Bogdanov, P., Corraliza, L., et al. (2016) [56]	Db/db	Retina	ERG Immunohistochemistry TUNEL assay Albumin leakage monitoring Western blot	None	10 weeks of age	Mice retinas showed higher percentage of apoptotic cells in whole retina and Outer nuclear layer (ONL), Inner nuclear layer (INL), and Ganglion Cell (GC) layer, overexpression of VEGF and IL-1β, increased blood-retinal barrier (BRB) breakdown, glial activation, levels of TUNEL-positive cells, and lower amplitudes in a- and b-waves compared to controls.
Ly, A., Scheerer, M.F., Zukunft, S., et al. (2014) [61]	Db/db	Retina	Immunofluorescence RT-qPCR	None	10 weeks of age	Mice retinas showed decreased levels of proteins related to synaptic transmission and cell signaling, increased VGLUT1, and decreased Slc17a7/VGLUT1 compared to controls.
Sohn, E.H., Van Dijk, H.W., Jiao, C., et al. (2016) [41]	Db/db STZ	Retina	SD-OCT	Yes	10 weeks of age	Mice retinas showed thinning in NFL and GCL compared to controls.
Tadayoni, R., Paques, M., Gaudric, A., et al. (2003) [62]	Db/db	Retina	Modified epifluorescence microscopy FITC-labeled RBC Rhodamine-labeled WBC	Yes	10 weeks of age	Mice retinas showed higher RBC velocity in retinal capillaries compared to controls.
Byrne, E.M., Llorián-Salvador, M., Lyons, T.J., et al. (2021) [57]	Db/db	Retina	Albumin and Isolectin B4 staining pJAK1 staining	None	12 weeks of age	Mice retinas showed BRB leakage and elevated pJAK1 levels compared to controls.
Jung, E., Kim, J., Kim, C.S., et al. (2015) [63]	Db/db Oxygen-induced retinopathy	Retina	Immunohistochemistry TUNEL and FITC-dextran leakage	None	12 weeks of age	Mice retinas showed increased E/P ratio, TUNEL-positive pericytes, and endothelial cells compared to controls.
Tang, L., Zhang, Y., Jiang, Y., et al. (2011) [64]	Db/db	Retina	Western blot Fluorometric assay Light microscopy	None	14 weeks of age	Mice retinas showed thinner whole central retina, INL, photoreceptor layer, decreased GC, elevated protein expression of ER stress biomarkers, BiP, PERK, ATF6, and active caspase-12 compared to controls.
Yang, Q., Xu, Y., Xie, P., et al. (2015) [65]	Db/db	Retina	Pattern electroretinogram (PERG) Fundus fluorescein angiography (FFA) OCT Immunohistochemistry	Yes	20 weeks of age	Mice retinas showed reduced P1 amplitudes and increased TUNEL-positive cells localized in GCL and INL compared to controls.
Liu, M., Pan, Q., Chen, Y., et al. (2015) [66]	Db/db	Retina	PAS staining Immunofluorescent staining Light microscopy	None	20 weeks of age	Mice retinas showed reduced whole retinal thickness, increased acellular capillaries, increased CAS-3 expression in photoreceptor layer, and increased MMP-2 and -9 mRNA expression compared to controls.
Clements, R.S. Jr., Robison, W.G. Jr., Cohen, M.P. (1998) [67]	Db/db	Retina	Histologic analysis Morphometric analysis	None	22 weeks of age	Mice retinas showed microvessel basement membrane thickening and 20% greater fractional basement membrane area compared to controls.
Sampedro, J., Bogdanov, P., Ramos, H., et al. (2019) [58]	Db/db	Retina	ERG in vivo TUNEL Evans blue Immunofluorescence Western Blot RT-PCR	None	24 weeks of age	Mice retinas showed lower amplitude a- and b-waves, aberrant GFAP extent, increased TUNEL-positive cells in INL and ONL, decreased total cells in GCL, INL, and ONL, upregulation of several proinflammatory cytokines (mRNA and protein), including IL-1β and IL-6, elevated TNFα, upregulation of NLRP3 inflammasome pathway, overexpression of IL-18, albumin leakage, extravasation of Evans Blue, and elevated protein levels of VEGF compared to controls.
Barile, G.R., Pachydaki, S.I., Tari, S.R., et al. (2005) [68]	Db/db	Retina	ERG Immunohistochemistry, RT-qPCR, autofluorescence, ELISA	None	24 weeks of age	Mice retinas showed prolonged oscillatory potentials and b-waves, accelerated development of acellular capillaries and pericyte ghosts, and increased AGE accumulation in retinal extracellular matrix compared to controls.
Hinder, L.M., Sas, K.M., O’Brien, P.D., et al. (2019) [69]	Db/db	Retina	ELISA Virtual-reality optokinetic tracking	None	24 weeks of age	Mice retinas showed features of early-stage DR, increased retinal apoptosis, and compromised visual performance compared to controls.
Hammer, S.S., Beli, E., Kady, N., et al. (2017) [70]	Db/db	Retina	Cell culture RT-qPCR Oxysterol measurement BMDM siRNA transfection Western blot ELISA	None	40 weeks of age	Mice retinas showed increased acellular capillaries, expression of TNFα, decreased LXR-SIRT signaling, levels of LXRα, Sirtuin 1, LXR, and CAC migration compared to controls.
Beli, E., Yan, Y., Moldovan, L., et al. (2018) [71]	Db/db	Retina	Trypsin digest Immunofluorescence staining qRT-PCR	Yes	44 weeks of age	Mice retinas showed increased numbers of acellular capillaries, IBA-1 positive cells, and CD45+ cell infiltration compared to controls.
Kern, T.S., Engerman, R.L. (1996) [72]	Db/db	Retina	Trypsin digest	Yes	60 weeks of age	Mice retinas showed increased acellular capillaries compared to controls.
Cheung, A.K., Fung, M.K., Lo, A.C., et al. (2005) [73]	Db/db	Retina	Immunohistochemistry Retinal sections Digital photographing	None	60 weeks of age	Mice retinas showed increased AR-immunoreactivity, expression of AR levels, formation of nitrotyrosine, GFAP Muller cell processes in Inner limiting membrane (ILM), Inner plexiform layer (IPL), outer plexiform layer (OPL), PCNA-positive cells, IgG extravasation from capillaries, caspase-3 activity, and >25% decreased pericyte density compared to controls.

Once outcomes such as visual acuity decreases, reduced contrast sensitivity, ERG changes, NFL-GCL thinning, RGC and amacrine cell neurodegeneration, and gliosis have developed, they persist and intensify with the duration of DM, with the exception of leukostasis and inflammation [17]. Although a variety of markers have been used throughout the literature to study the role of inflammation in DR, some of the more common markers included the elevated number of leukocytes, leukocyte adhesion, microglial infiltration, production of cytokines, eNOS, and several caspases [20,32,33,38,43,44,45]. Every murine model of DR shows the progression of the disease, i.e., the extent and nature of the damage of the retina increases, but not the transition from NPDR to PDR.

### 2.3. Rat Model of Type-1 Diabetic-Induced Retinopathy

The rat model has also been in use for some time by researchers and has been shown to reproduce a similar pathology to human DM. Most commonly, STZ injections have been used to induce a non-autoimmune form of T1D. The rat model has also shown a delay in the detection of diabetic retinopathy after induction/onset of diabetes; however, it may be less than that seen in T1D mice. In contrast to the STZ-induced mice, the STZ-induced DM rats demonstrated leakage of retinal blood vessels to very small molecules such as fluorescein after as early as 2 days of DM, as can be seen in Table 3 [12,14,74]. A shorter delay in the T1D rats may suggest that they are less appropriate models for studying resilience to DR when compared to T1D mouse models. However, this conclusion is weak because different outcomes were performed in different labs to assess vascular damage.

### 2.4. Potential Pathways of Resilience to DR-Associated Damage

Resilience is likely to prevent those events that cause damage to the retina. These would include oxidative stress and cytokines [39,47,48,49,86,87,88,89,90,91,92,93,94]. Also, engaging endogenous systems, such as mitophagy, which prevent mitochondrial oxidative stress, are likely to be at least part of the mechanism of resilience [7].

## 3. Discussion

DR is a complex and progressive disease characterized by a spectrum of pathologic changes, which include retinal vascular abnormalities, inflammatory processes, and neurodegeneration. The delay between the onset of DM and the manifestation of retinal damage/dysfunction indicates the existence of resistance to DR, which must be overcome before outcomes of the disease appear. 

There are multiple explanations for resilience, i.e., the absence of damage in the face of insults that are capable of causing such damage. It is possible that resilience relates to the level of sensitivity of current approaches to detect damage. Damage may commence with the onset of DM, but its extent is below the level of detection. This scenario predicts that lowering the detection limit will reduce or eliminate the period of resilience. However, technological advances that have been made to date have not demonstrated that DR manifests coincident with the onset of DM.

An alternative explanation of resilience is that cells adapt to hyperglycemia (HG) and, thereby, become resistant to its deleterious effects. Our recent publications provided compelling support for this possibility. In DM mice, resistance is transient, and its loss sets the stage for accumulation of damage, i.e., manifestation of DR. More specifically, the retinal vasculature of mice that had experienced a short duration of DM acquired resistance to oxidative stress/ischemia- or cytokine-induced death [7]. As the duration of DM was prolonged, protection waned and was replaced by increased vulnerability; i.e., there was more insult-driven death in retinal vessels from DM mice as compared with age-matched non-DM mice [7]. Finally, the appearance of vulnerability coincided with the manifestation of DR—vascular and neural dysfunctions [7].

The underlying mechanism of resilience involves increased mitophagy (clearance of dysfunctional mitochondria), which boosts mitochondrial functionality and prevents mitochondrial oxidative stress in the face of HG [6]. It resides in endothelial cells but not in pericytes of human retinal capillaries [6]. Whether resilience exists in other retinal cell types has not been investigated to date. The antioxidant component of resilience has been noted by other groups reporting that expression of RBP3 (retinol-binding protein 3), an antioxidant [89,90,91], is elevated in patients who are resistant to diabetic retinopathy [92,93]. Taken together, it appears that retinopathy is held in check during the resilience phase because the endothelium has engaged in mitophagy-based defense against the deleterious effects of HG (Figure 2). Loss of resilience is a prerequisite for accumulation of damage in the retina, which is recognized as DR.

The appreciation of resilience, or the innate ability of the retina to remain healthy in the face of DM, raises essential questions regarding the potential role of genetic susceptibility and genetic protection in the development of DR. Moreover, it opens the door to a world of novel biomarkers and potential therapeutic targets of DR that must be further explored to manage people at risk of developing DR and resisting the deleterious effects of DM. Such alternatives are necessary because the current prophylactic options are not effective for all people. This is especially crucial when considering that 26% of the diabetic US population has some form of DR [94].

## Figures and Tables

**Figure 2 ijms-25-01610-f002:**
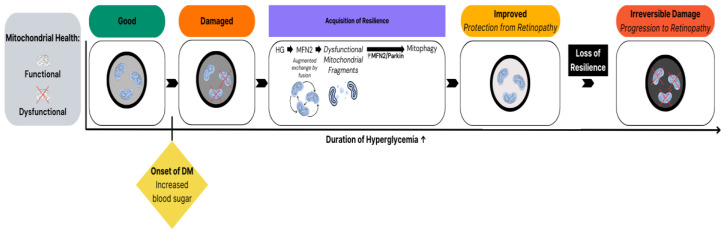
Mitophagy is an essential component of resilience. The onset of DM, i.e., enduring elevation of blood sugar, triggers mitochondrial adaptation, which includes enhanced clearance of dysfunctional mitochondria. Deterioration of this innate defense mechanism results in self-perpetuating progress damage of the retina, which manifests as DR.

**Table 3 ijms-25-01610-t003:** Articles of note discussing Type-1 diabetic rat models.

Authors, (Year)	Model	Collected Tissue	Mechanism of Analysis in Diabetic Retinopathy	Monitoring of DR Progression	Earliest Detection of Diabetic Retinopathy	Result/Outcome
Jones, C.W., Cunha-Vaz, J.G., Rusin, M.M. (1982) [14]	STZ Alloxan	Retina	Vitreous fluorophotometry	None	2 days of diabetes	Streptozocin- or alloxan-induced diabetic rat retina showed increases in vitreal fluorescein levels compared to control rats.
Miyamoto, K., Khosrof, S., Bursell, S.E., et al. (1999) [75]	STZ	Retina	AOLF and Fluorescein Angiography Isotope Dilution Technique Ribonuclease protection assay	Yes	1 week of diabetes	Rat retina showed increased albumin permeation compared to controls.
Park, S.H., Park, J.W., Park, S.J., et al. (2003) [76]	STZ	Retina	Immunohistochemistry Histologic analysis Electron microscopy TUNEL	Yes	1 week of diabetes	Rat retina showed myelinated and multivesicular features of mitochondria, increased thickness of inner retina, and the vitreal radial processes of Müller cells stained moderately and became prominent compared to controls.
Ishida, S., Usui, T., Yamashiro, K., et al. (2003) [77]	STZ	Retina	FITC FITC-dextran ELISA Immunohistochemistry	Yes	2 weeks of diabetes	Rat retina showed increased leukostasis and blood-retinal barrier (BRB) breakdown compared to controls.
Rungger-Brändle, E., Dosso, A.A., Leuenberger, P.M. (2000) [78]	STZ	Retina	Immunofluorescence Confocal microscopy Western blot Evans Blue staining	Yes	2 weeks of diabetes	Rat retina showed leakage of BRB compared to controls.
Barber, A.J., Lieth, E., Khin, S.A., et al. (1998) [79]	STZ	Retina	TUNEL Nissl staining Microscopy	Yes	4 weeks of diabetes	Rat retina showed increased TUNEL-positive cells, suggesting retinal cell apoptosis compared to controls.
Aizu, Y., Oyanagi, K., Hu, J., et al. (2002) [80]	STZ	Retina	ERG Electron microscopy Optical microscopy	Yes	4 weeks of diabetes	Rat retina showed reduced thickness of inner plexiform layer (IPL, and photoreceptor segment layer (PSL), reduced a- and b-waves, oscillatory potentials, and deepened hollows in basal infoldings of retinal pigment epithelium (RPE) compared to controls.
Barber, A.J., Antonetti, D.A., Gardner, T.W. (2000) [81]	STZ	Retina	Immunofluorescence Histochemistry	Yes	8 weeks of diabetes	Rat retina showed astrocytes with reduced GFAP intensity, multiple processes, enlarged cell bodies, and hypertrophied compared to controls. They also demonstrated regions of heavily punctate immunoreactivity and absence of distinct junctional distribution in major arterioles compared to controls.
Fico, E., Rosso, P., Triaca, V., et al. (2022) [82]	STZ	Retina	Microscopy Immunofluorescence histology Western blot Confocal imaging	None	8 weeks of diabetes	Rat retina showed fragmentation of retinal vessels, high expression Rock1 level, proNGF and its glycosylated form, decreased pVEGFR2 expression, and Akt levels compared to controls.
Gong, C.Y., Lu, B., Hu, Q.W., et al. (2013) [83]	STZ	Retina	Immunofluorescence RT-PCR	Yes	12 weeks of diabetes	Rat retina showed morphological changes in INL, ONL, increased number of new vessels, mRNA expression of VEGF, VEGFR1, and VEGFR2 compared to controls.
Lieth, E., Barber, A.J., Xu, B., et al. (1998) [84]	STZ	Retina	ELISA Immunohistochemistry Immunoblotting glutamate-to-glutamate conversion	Yes	12 weeks of diabetes	Rat retina showed increased GFAP IR and increased glutamate levels compared to controls.
Agardh, E., Bruun, A., Agardh, C.D. (2001) [85]	STZ	Retina	Immunohistochemistry	Yes	24 weeks of diabetes	Rat retina showed increased GFAP and glial cell immunoreactivity, and horizontal cell numbers with decreased branching of terminals compared to controls.

Once outcomes such as visual acuity decreases, reduced contrast sensitivity, ERG changes, NFL-GCL thinning, RGC and amacrine cell neurodegeneration, and gliosis have developed, they persist and intensify with the duration of DM, with the exception of leukostasis and inflammation [17]. Although a variety of markers have been used throughout the literature to study the role of inflammation in DR, some of the more common markers included the elevated number of leukocytes, leukocyte adhesion, microglial infiltration, production of cytokines, eNOS, and several caspases [20,32,33,38,43,44,45]. Every murine model of DR shows the progression of the disease, i.e., the extent and nature of the damage of the retina increases, but not the transition from NPDR to PDR.

## Data Availability

Not applicable.

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
