# Peer review of "Manifestation of Pathology in Animal Models of Diabetic Retinopathy Is Delayed from the Onset of Diabetes"

_ijms, 2024, doi:10.3390/ijms25031610_

Round 1

Reviewer 1 Report

Comments and Suggestions for Authors

Cubillos et al. present a review focusing on the relationship between the manifestation of retinal pathology and the onset of diabetes (diabetes mellitus, DM) in animal models of diabetic retinopathy (DR).

The manuscript is well written, and the message is presented clearly.

The authors conclude that DR does not develop coincident with the onset of DM in murine and rat models, and that the progressive nature of DR suggests the existence of a resilience mechanism. This has important implications for investigation of novel therapies preventing DR occurrence and progression.

The manuscript brings together and discusses many DR models utilized in the studies and presents an added value to the field.

Author Response

We thank this reviewer for their positive evaluation of our manuscript. 

Reviewer 2 Report

Comments and Suggestions for Authors

The authors have done a great job in gathering information regarding DR rodent mouse models in this review. Although the evidence they present supports their hypothesis regarding a failure of resilience against DR-associated damage may lead to DR tissue damage in humans based on these rodent studies, there are sections where the authors can expand in their review. My comments can be found below:

Introduction

The authors could include a description of the common pathologic features of DR rodent models and how these features are experimentally detected/studied. This would help to interpret and understand Tables 1-3. 

Are there any major problems regarding DR rodent models versus human DR? It might be worthwhile to discuss them in the introduction. 

Are there any apparent advantages between using mice versus rodents? This may be worth discussing as well. 

[Lines 47-55]: Can the authors provide more details to this paragraph? Which T1D and T2D models? Which antioxidant genes are changed or at least which antioxidant transcriptional pathway are affected in these animals? 

Can the authors generate a schematic figure from their hypothesis described in [Lines 91-94]?

I would consider moving [Lines 56-77] to section 2.1 since this section refers to DR in T1D mouse models.

I would consider moving [Lines 78-87] to section 2.2 since this section refers to DR in T2D mouse models.

Literature Review

The authors have done a wonderful job summarizing previous studies in Tables 1-3. However, it would be nice to have a column in these tables that describes the mouse model (i.e. STZ-induced or InsAkita). This column would make these tables more effective. 

The authors could also discuss the information in Tables 1-3 in paragraphs that enforce statements in these sections. For instance, “Once these outcomes have developed, they persist and intensify as DM and DR continues to progress, with the exception of leukostasis and inflammation [16].” What models show pathologies worsening with progression? What pathologies are most affected? What features of leukostasis and inflammation were examined? Are these consistent across models? Comparing findings from these multiple DR models would greatly benefit this review which readers may appreciate. 

It would be nice for the authors to speculate on what pathways may be important for the resilience to DR-associated damage in these models, and this could be used to generate a Section 2.4. Based on the introduction, it appears that quenching oxidative stress is important for this resilience. I would also assume this would include mitochondria as well since reactive oxygen species are predominantly generated at mitochondria. This information would be nice for this review as it supports the author’s hypothesis about DR formation.

Author Response

Introduction 

The authors could include a description of the common pathologic features of DR rodent models and how these features are experimentally detected/studied. This would help to interpret and understand Tables 1-3. 

As requested, we modified the Introduction of the manuscript to include a description of the common pathological features of DR rodent models and how these features are studied.

Are there any major problems regarding DR rodent models versus human DR? It might be worthwhile to discuss them in the introduction. 

Yes; animal models of DR do not progress to the proliferative form of DR, which is characterized by neovascularization of the retinal vessels.  Retinal neovascularization can be modeled with the oxygen-induced retinopathy protocol, however the animals are not DM in this experimental setting, and hence constitute a major difference from the clinical scenario.  We include this information in the expanded version of the description of common pathological features of DR rodent models, which we outlined in our response to the previous comment.

Are there any apparent advantages between using mice versus rodents? This may be worth discussing as well. 

As requested, we added a paragraph to the Introduction to discuss the advantages of mice (availability of genetically modified mice, cost, commonly used) and rats (size of the eye). 

[Lines 47-55]: Can the authors provide more details to this paragraph? Which T1D and T2D models? Which antioxidant genes are changed or at least which antioxidant transcriptional pathway are affected in these animals? 

The requested information has been added to this paragraph.  

Can the authors generate a schematic figure from their hypothesis described in [Lines 91-94]?

We include a schematic figure (Figure 1) to illustrate this hypothesis. 

I would consider moving [Lines 56-77] to section 2.1 since this section refers to DR in T1D mouse models.

The request change has been made. 

I would consider moving [Lines 78-87] to section 2.2 since this section refers to DR in T2D mouse models.

The request change has been made. 

Literature Review

The authors have done a wonderful job summarizing previous studies in Tables 1-3. However, it would be nice to have a column in these tables that describes the mouse model (i.e. STZ-induced or InsAkita). This column would make these tables more effective. 

A column describing the mouse model has been added to each of the tables. 

The authors could also discuss the information in Tables 1-3 in paragraphs that enforce statements in these sections. For instance, “Once these outcomes have developed, they persist and intensify as DM and DR continues to progress, with the exception of leukostasis and inflammation [16].” What models show pathologies worsening with progression?? What features of leukostasis and inflammation were examined? Are these consistent across models? Comparing findings from these multiple DR models would greatly benefit this review which readers may appreciate. 

We modified the manuscript to include a paragraph (at the end of each Table) that enforces the statements in these sections. 

It would be nice for the authors to speculate on what pathways may be important for the resilience to DR-associated damage in these models, and this could be used to generate a Section 2.4. Based on the introduction, it appears that quenching oxidative stress is important for this resilience. I would also assume this would include mitochondria as well since reactive oxygen species are predominantly generated at mitochondria. This information would be nice for this review as it supports the author’s hypothesis about DR formation.

As requested, we generated section 2.4 in which we speculate on the nature of the pathways that are important for resilience to DR-associated damage.  Furthermore, we added Figures 1 and 2 that illustrate various aspects of the key concepts.

Reviewer 3 Report

Comments and Suggestions for Authors

The manuscript titled "Manifestation of pathology in animal models of diabetic retinopathy is delayed from the onset of diabetes" is an interesting and valuable review. It analyses and explores existing research on the pathogenesis of diabetic retinopathy (DR) in animal models, providing valuable insights into the topic.

The tables should include information on the tissues or body fluids where the researched DR marker was identified. Additionally, details on whether the progression of DR was monitored in the researches, if other micro- or macrovascular complications of diabetes mellitus were identified in any of the studies, and the lifespan of the researched animals should also be incorporated.

Given the significance of the topic, an expansion of the discussion section is needed. Beyond the presentation of relevant research, the obtained results should be interconnected and thoroughly commented upon. Building upon this analysis, the discussion should propose potential directions for further research on the pathogenesis and diagnosis of DR in humans, along with exploring potential therapeutic possibilities.

The claim, "It is possible that DM-induced damage begins immediately upon elevation of blood sugar, but such damage is below the detection limit of the current approaches to detect it," should be substantiated and supported by existing knowledge, establishing the groundwork for this hypothesis.

Furthermore, the discussion should delve into the potential role of genetic susceptibility and genetic protection in the development of DR. It is essential to explore potential new biomarkers of DR and specify potential therapeutic targets to provide a comprehensive overview of the topic.

Author Response

The tables should include information on the tissues or body fluids where the researched DR marker was identified. Additionally, details on whether the progression of DR was monitored in the researches, if other micro- or macrovascular complications of diabetes mellitus were identified in any of the studies, and the lifespan of the researched animals should also be incorporated.

We added the requested information, which was routinely included in the publications, to the tables and/or paragraph below the tables.  This was only the tissue where the DR marker was identified (retina) and whether a given outcome progressed as the duration of DM was prolonged.  The publications did not contain information regarding other complication or the lifespan of the animals, and hence we were not able to add this information. 

Given the significance of the topic, an expansion of the discussion section is needed. Beyond the presentation of relevant research, the obtained results should be interconnected and thoroughly commented upon. Building upon this analysis, the discussion should propose potential directions for further research on the pathogenesis and diagnosis of DR in humans, along with exploring potential therapeutic possibilities.

As requested, we added the last paragraph of the Discussion.  

The claim, "It is possible that DM-induced damage begins immediately upon elevation of blood sugar, but such damage is below the detection limit of the current approaches to detect it," should be substantiated and supported by existing knowledge, establishing the groundwork for this hypothesis.

As requested, we added the second, third and fourth paragraphs of the Discussion. In addition, we added Figures 1 and 2, and generated section 2.4.  

Furthermore, the discussion should delve into the potential role of genetic susceptibility and genetic protection in the development of DR. It is essential to explore potential new biomarkers of DR and specify potential therapeutic targets to provide a comprehensive overview of the topic.

As requested, we added the last paragraph of the Discussion.

Round 2

Reviewer 2 Report

Comments and Suggestions for Authors

The authors have addressed my concerns and have made substantial revisions to their manuscript. 

Reviewer 3 Report

Comments and Suggestions for Authors

The authors have responded appropriately to the suggestions and have implemented all the requested changes, resulting in an improved quality of the manuscript. Therefore, the manuscript can be accepted in its current form.